# Optimizing Markov Chain Monte Carlo Convergence with Normalizing Flows and Gibbs Sampling

**Christoph Schönle, Marylou Gabrié**
CMAP, CNRS, École polytechnique,
Institut Polytechnique de Paris, 91120 Palaiseau, France
`{christoph-martin.schonle,marylou.gabrie}@polytechnique.edu`

## Abstract

Generative models have started to integrate into the scientific computing toolkit. One notable instance of this integration is the utilization of normalizing flows (NF) in the development of sampling and variational inference algorithms. This work introduces a novel algorithm, GflowMC, which relies on a Metropolis-within-Gibbs framework within the latent space of NFs. This approach addresses the challenge of vanishing acceptance probabilities often encountered when using NF-generated independent proposals, while retaining non-local updates, enhancing its suitability for sampling multi-modal distributions. We assess GflowMC's performance concentrating on the $\phi^4$ model from statistical mechanics. Our results demonstrate that by identifying an optimal size for partial updates, convergence of the Markov Chain Monte Carlo (MCMC) can be achieved faster than with full updates. Additionally, we explore the adaptability of GflowMC for biasing proposals towards increasing the update frequency of critical coordinates, such as coordinates highly correlated to mode switching in multi-modal targets.

## 1 Introduction

The potential unlocked by deep learning in high-dimensional function approximation holds great promise for scientific computing. Noteworthy achievements include the development of machine learning force fields (e.g., (1)) and Physics Informed Neural Networks (e.g., (2)), both of which exemplify the remarkable capabilities of deep learning in predictive tasks. However, the impact of deep learning extends beyond mere prediction, as deep generative models have also found a wide spectrum of applications in scientific computing; from the creation of interpretable generative models (e.g., contact predictions derived from Boltzmann machine learning (3)) to the acquisition of dimension-reduced representations (e.g., (4)). Moreover, generative models have demonstrated their effectiveness in enhancing Monte Carlo methods, a primary focus of the present study.

Generative models featuring tractable likelihoods and straightforward sampling mechanisms, such as normalizing flows (NF) and autoregressive models (ARM), can greatly facilitate the inference process for high-dimensional distributions. Once the generative model has been trained to directly approximate the target measure - for instance using variational inference (VI) (5; 6) - it can be used as an adaptive proposal distribution in Monte Carlo algorithms such as importance sampling (7; 8) or Metropolis-Hastings Markov chains (9; 10). Promising results have been obtained in moderately high-dimensions, yet scalability challenges are becoming evident as the quality of the target measure approximation drops with increasing dimension (11; 12).

While using a learned model to generate independent configurations is too challenging for highly complex or very large systems, the possibility to leverage deep generative models to propose *local* or *partial* state updates in Markov Chain Monte Carlos (MCMCs) has also been explored (13; 14; 15; 16; 17; 18). In the present work, we propose a novel strategy in this direction, Gibbs-Flow Monte Carlo

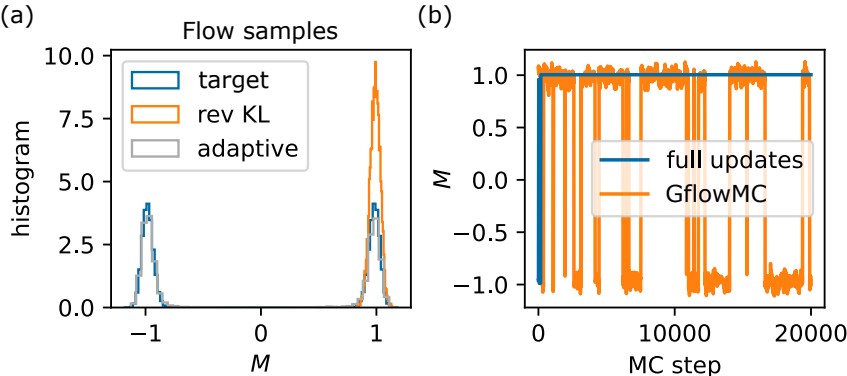

Figure 1: **The challenges of using normalizing flows for MCMC sampling of multimodal target distributions in high-dimensions on the example of the 2d $\phi^4$ model below the critical temperature.** (a) Target distribution of magnetisations and samples from two flows trained with the reverse KL divergence as the objective and in the adaptive fashion by (19). The former flow suffers from the well-known mode collapse. The latter flow was used for MCMC sampling in (b). Even when the flow model covers well the two modes, chains with full update proposals suffer from long streaks of rejection and can remain stuck in a single mode. GflowMC chains with partial updates in latent space do not. The plots shown are for $L = 20$, and $\theta = 1.6$ in eq. (4). GflowMC was ran with roughly optimal partial size, $d_u = 50$. Out of 500 arbitrarily initialized chains in each case, the worst chain with the least mode switches is shown.

(GflowMC), a Metropolis-within-Gibbs sampler implemented in the pull-back of a target distribution through an NF's transport map. Using partial updates in the *latent* space, GflowMC improves mixing speed by finding the sweet spot between the amount of state refresh in the proposals and the rate of the proposals' acceptance while keeping the ability to mix between modes in multi-modal targets.

## 2 Gibbs Flow Monte Carlo (GflowMC)

GflowMC is a sampling algorithm for distributions admitting a density with respect to the Lebesgue measure on $\mathbb{R}^d$; this target density is denoted $\rho^*$.

**NF-preconditioned sampling** An NF model on $\mathbb{R}^d$ combines a simple base distribution on the same space, with density denoted here by $\nu_{\mathrm{B}}$, and a learnable diffeomorphism $T_\alpha$, thereby defining a push-forward distribution with density at $x \in \mathbb{R}^d$

$$\rho_\alpha(x) = \nu_{\mathrm{B}} \left( T_\alpha^{-1}(x) \right) |\det \nabla_x T_\alpha^{-1}(x)|, \tag{1}$$

following the change of variable formula. Different strategies have been developed to parametrize $T_\alpha$ to be easily invertible and ease the computation of the Jacobian $\nabla_x T_\alpha^{-1}(x)$, we refer the reader to the reviews (20; 21). The proposed algorithm is agnostic to the specific parametrization strategy. The pull-back of the target distribution admits the density

$$\nu_\alpha^*(z) = \rho^* \left( T_\alpha(z) \right) |\det \nabla_z T_\alpha(z)|. \tag{2}$$

We will assume that the NF's parameters have been adjusted such that $T_\alpha$ approximately transports $\nu_{\mathrm{B}}$ to $\rho^*$ (see (5; 7; 19) for training schemes). We expect $\rho_\alpha \approx \rho^*$ in the *direct* space and $\nu_{\mathrm{B}} \approx \nu_\alpha^*$ in the *latent* space. As $\nu_{\mathrm{B}}$ is deliberately selected for its ease of sampling, several works have presented sampling strategies for $\rho^*$ by addressing the equivalent task of sampling from $\nu_\alpha^*$ and subsequently applying the transport map $T_\alpha$ (13; 14; 15). While these prior works employed gradient-based *local* samplers within the latent space (as discussed in related works), GflowMC takes a distinct approach by employing *partial* updates in the latent space.

**Metropolis-within-Gibbs on the pull-back** When it is possible to evaluate and sample conditional distributions for subsets of coordinates from the base distribution $\nu_{\mathrm{B}}$, an efficient Metropolis-within-Gibbs sampling approach for the pull-back $\nu_\alpha^*$ can be devised. Namely, GflowMC is an MCMC using

---
**Algorithm 1:** Gibbs Flow Monte Carlo with size $d_\mathrm{u} \in [\![1, d]\!]$ updates

**inputs :** Target density $\rho^*$, Trained flow $T_\alpha$, Indices' weights $w \in \Delta^{d-1}$, Starting point $x_0$

**1** $z_0 = T_\alpha^{-1}(x_0)$                 `// compute the starting point in latent space`

**2 for** $t = 1 \cdots T$ **do**

**3**     $S = (i_1, \cdots i_{d_\mathrm{u}}) \sim H(w)$             `// select subset to update`

**4**     $z_S' \sim \nu_\mathrm{B}(z_S'|z_{\backslash S}^t)$             `// Gibbs partial refresh proposal`

**5**     $z_S^{t+1} = z_{\backslash S}' = z_{\backslash S}^t$

**6**     Draw $u \sim \mathcal{U}([0, 1])$              `// Metropolis-Hastings accept-reject`

**7**     **if** $u < \min(1, \gamma(z'|z))$ **then**

**8**        $z_S^{t+1} = z_S'$

**output :** $(x^1, \cdots, x^T) = (T_\alpha(z^1), \cdots, T_\alpha(z^T))$

---

partial updates in latent space. The full procedure is described in algorithm 1 using the notations introduced hereafter.

Let $w = (w_i)_{i=1}^d \in \Delta^{d-1} \subset \mathbb{R}^d$ represent a normalized vector of weights with strictly positive entries within the simplex $\Delta^{d-1}$; it defines a probability distribution over coordinate indices. Additionally, let $S \subset [\![1, d]\!]$ denote a subset of dimension indices with a cardinality of $|S| = d_\mathrm{u}$, and let $\backslash S = S^c$ denote its complementary set. Finally, define $z_S = (z_i)_{i \in S} \in \mathbb{R}^{d_\mathrm{u}}$ and $z_{\backslash S} = (z_i)_{i \notin S} \in \mathbb{R}^{d-d_\mathrm{u}}$ as the corresponding vectors. At iteration $t + 1$ of the Markov Chain, a subset $S$ is selected by drawing $d_\mathrm{u}$ indices from the non central hypergeometric distribution with weights $w$, here noted $H(w)$ (i.e. drawing from the biased multinomial distribution without replacement). A proposed update of $z_S^t$ is sampled from the conditional base distribution $z_S' \sim \nu_\mathrm{B}(z_S'|z_{\backslash S}^t)$ and accepted with the Metropolis-Hastings probability:

$$\gamma(z_S'|z^t) = \frac{\nu_\mathrm{B}(z_S'|z_{\backslash S}^t)}{\nu_\mathrm{B}(z_S^t|z_{\backslash S}^t)} \frac{\nu_\alpha^*(z^t)}{\nu_\alpha^*(z')}. \tag{3}$$

Given a factorized base density, $z_S$ becomes independent of $z_{\backslash S}$ and the conditional $\nu_\mathrm{B}(z_S|z_{\backslash S})$ equals the marginal $\nu_\mathrm{B}(z_S)$. It occurs for the most common choice of $\nu_\mathrm{B}$, the standard Gaussian, where evaluating and sampling from $\nu_\mathrm{B}(z_S)$ is straightforward for subsets $S$ of any size.

GlowMC is an instance of a "random scan" Metropolis-within-Gibbs sampler. A proof of convergence is given for instance by (22), and (23; 24) discuss the choice of the scan, that is the coordinate selection scheme here encapsulated by the weight vector $w$.

**Related works and motivation**    The application of normalizing flows for MCMC sampling has multiple challenges. Notably, for multimodal targets, training without data in the energy-based approach suffers from mode collapse (see fig. 1(a)) (25; 26). A recent approach was proposed to quantify and detect this for a trained flow (27). With the adaptive strategy introduced by (19), the problem is alleviated, but even with a flow covering all modes, sampling from it can be problematic. A popular choice is the independent Metropolis-Hastings (IMH) sampler (9; 7; 10; 28; 19), which is the limiting case of GflowMC for $d_\mathrm{u} = d$. However, independent proposal samplers are known to be nearly impossible to tune in high-dimension. Even proposals parametrized by highly expressive deep neural networks will eventually suffer from a curse of dimensionality, bringing drowning acceptance rates, as exemplified on the statistical mechanics $\phi^4$ model by (11) and also shown in fig. 1(b).

Gibbs samplers can circumvent this vanishing acceptance by directly sampling from tractable conditional distributions. Yet, the strategy is limited to updating a couple of degrees of freedom at the time due to the difficulty of manipulating the conditionals in the general case. Alternatively, a Metropolis-within-Gibbs strategy allows to update sets of coordinates of any size following the proposal-rejection logic. The size of the updates can be tuned to optimize the convergence (29; 30), as we discuss for GflowMC in the next section, and akin to the tuning of the step size in Langevin samplers (31). In computational physics, "cluster" algorithms consist in updating simultaneously groups of spatially adjacent degrees of freedom and can greatly improve performance compared to purely local MCMCs (see e.g. (32)).

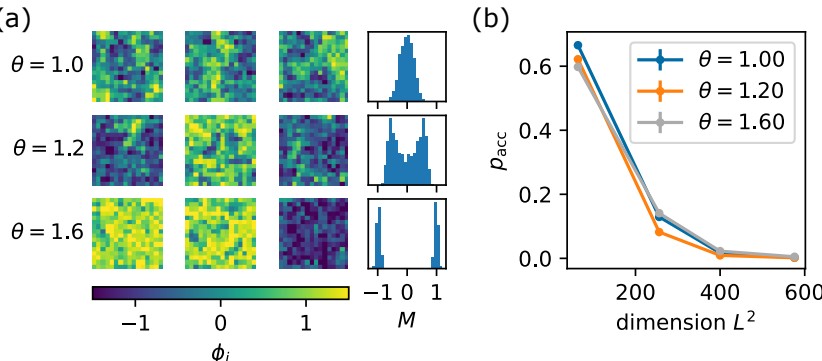

Figure 2: (a) Typical samples and distribution of the magnetisation $M$ of the $\phi^4$ model at lattice size $L = 16$. A phase transition from uni- to bimodal occurs around temperature $\theta = 1.2$ (35). (b) Using normalizing flows in an IMH sampler (full updates), the acceptance rate quickly vanishes as dimension increases.

Inspired by cluster updates, (17) proposed to use ARMs as proposals in a Metropolis-within-Gibbs sampler in coordinate space. ARMs are suited for discrete distributions, such as for the spin-systems considered in the latter work. ARMs also rely on an ordering of the coordinates, such that tractable conditionals are limited to subsets of coordinates following the factorization order. As a result, the proposed method can only define an ergodic Markov Chain by relying on the translational invariance of the target field systems for which the learned ARM model can be shifted over space. Conversely, GflowMC handles continuous distributions and its ergodicity requires no symmetry or invariance in the target distribution as any subsets of latent dimensions can be updated.

Other works concerned with continuous distributions have proposed to take *local* rather than *partial* updates in the latent space. NeutraMCMC methods run Metropolis Adjusted Langevin (MALA) (13) or Hamiltonian Monte Carlo (14; 15) on the pull-back in latent space. However, such local updates have limited take on mode mixing for multi-modal targets, as shown in the detailed study of (33). Meanwhile, the gradient-free transport elliptical slice sampling (34) can mix between modes, but appears less efficient than NeutraMCMCs and IMH, again according to (33). Below, we show that few coordinates updates can be identified to be related to mode switching and exploited to promote mode-switches in GflowMC.

## 3 Numerical results

**2d $\phi^4$ model**  We study a traditional benchmark model of machine learning assisted samplers for physical systems, the 2d $\phi^4$ model. The target distribution is the thermal equilibrium Boltzmann distribution $\rho^*(\phi) \propto e^{-E(\phi)}$, with the Hamiltonian:

$$E(\phi) = \sum_{i,j=1}^{L} \left[ \left( 2 - \frac{\theta}{2} \right) \phi_{i,j}^2 + \frac{1}{4} \phi_{i,j}^4 - \phi_{i+1,j}\phi_{i,j} - \phi_{i,j+1}\phi_{i,j} \right], \tag{4}$$

defined on a 2d square lattice of size $d = L^2$, with a tunable parameter $\theta$ playing the role of a temperature, and with periodic boundary conditions. The model can be seen as a continuous version of the Ising model: it undergoes a phase transition from a unimodal to a bimodal phase when varying $\theta$, illustrated in fig. 2(a). It has served as a benchmark problem in numerous works as its dimension and properties can be varied with different choices of $\theta$ and $L$ (9; 36; 11; 37; 19; 25; 38; 26).

We trained NFs for lattice sizes between $L^2 = 8^2$ and $24^2$ and for parameter values $\theta$ across the transition point adjusting architecture with model size as described in appendix A. No prior knowledge about the mode symmetry $\rho^*(\phi) = \rho^*(-\phi)$ is built-in, since we have the general case in mind where relative weights of modes are not known a priori. Training was done in the adaptive fashion introduced by (19), precise settings are given in appendix A. We stress that the emphasis of this work is not on the design and training of the flow, but on utilizing it optimally for MC sampling when it has shortcomings due to insufficient expressivity or training time.

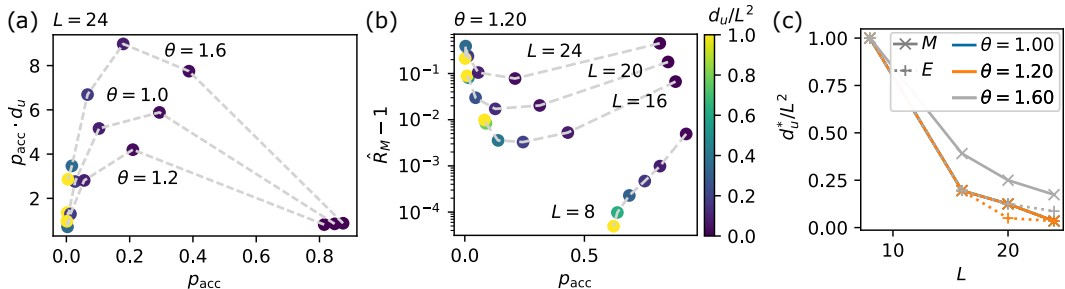

Figure 3: GlowMC convergence as a function of update subset size $d_{\mathrm{u}}$. (a) Effective update size of chains, and (b) GR statistics for the magnetisation, both shown against the varying acceptance rate. The color indicates $d_{\mathrm{u}}/L^2$, the fraction of latent sites for which updates are proposed. For each $L$, the left-most point with smallest acceptance always corresponds to full updates, the right-most point to single-site updates. (c): Optimal update fraction with increasing lattice size $L$. Data points in (a) an (b) were obtained as averaged over 4 independently trained flows, and these averages were used to determine each $d_u^*$ in (c).

**Metrics** To gauge the convergence of our MCMC chains, we studied two important observables, the energy $E$ and the magnetisation $M = \frac{1}{L^2} \sum_{i,j=1}^{L} \phi_{i,j}$, the field values averaged over the lattice. Since the energy $E$ is invariant with respect to a sign flip of the field configurations, it is ignorant about the two modes and as such a good indicator on how well the sampler works for within-mode exploration. In contrast, $M$, coined *order parameter* in the statistical mechanics literature, will only converge when the multi-modality has been accurately represented.

Different metrics exist to judge the quality of convergence for a given observable. One obvious choice is the integrated autocorrelation time, but it becomes increasingly unstable to estimate numerically with decreasing flow performance. Instead we focus here on the potential scale reduction statistics $\hat{R}$ from Gelman and Rubin (GR), which compares the variance of samples within each chain to the variance across chains (39; 40). Observe that $\hat{R} \geq 1$ by construction, perfectly converged chains would be signaled by $\hat{R} = 1$, and a typical stopping criteria is $\hat{R} \leq 1.01$. Unless specified otherwise, results here are presented for 500 parallel chains run for 20,000 MCMC steps.

**Optimal update size** Consistent with previous work (11), we show in fig. 2(b) that the average acceptance $p_{\mathrm{acc}}$ of full updates proposed by the trained flow drops quickly when increasing dimension $d = L^2$. Presumably this is due to the constant number of coupling blocks as well as increasing training time that would be required for comparable accuracy.

We ran MCMC chains with GflowMC for different update sizes $d_{\mathrm{u}}$, comparing the corresponding chains' convergence in fig. 3. Fig. Panel (a) reports a simple first metric, the effective number of updated sites per step, $p_{\mathrm{acc}} \cdot d_{\mathrm{u}}$, for the most difficult system size under consideration, $L = 24$, and three different values of $\theta$. Interestingly, there seems to be an approximately optimal acceptance rate at $p_{\mathrm{acc}} \approx 0.25$ which should be aimed for when deciding on the size of the partial updates. Evidence of this can also be seen in fig. 3(b) from the GR statistics of the magnetisation close to the critical temperature $\theta = 1.2$. Even though the overall quality of convergence decreases with system size (at equal length of MCMC chains), the optimal value of $p_{\mathrm{acc}}$ remains roughly constant across values of $L$ and $\theta$. The smallest system at $L = 8$ is an exception here: the acceptance rate of the flowMC with full updates at around 60% already surpasses the putative optimal value and thus any partial updating scheme will only slow down convergence. Analogous behaviour can be seen for the GR statistics of the energy $E$ (not shown). Panel (c) shows how the optimal fraction of latent coordinates updated, according the the optimal statistics of $E$ and $M$, decreases monotonously with $L$, manifesting the decreasing quality of the flow and increasing complexity of the sampling task. In light of this, the choice of $d_{\mathrm{u}}$ can be understood in the same way as in choosing the step size for a random walk or MALA Monte Carlo algorithm, for which an optimal target acceptance rate has been theoretically established in the case of simple target Gaussian distributions (31). Extensions of the previous work also derive an optimal acceptance rate for Metropolis-within-Gibbs algorithms (29; 30), and thus it is not surprising to find similar behaviour in our sampling scheme.

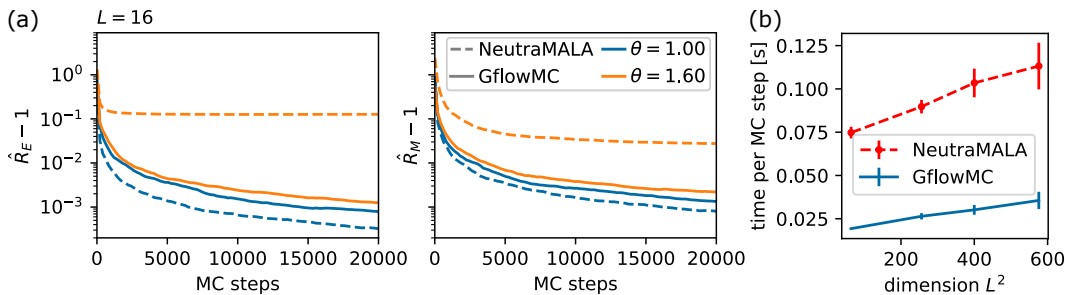

Figure 4: Comparison of samplers utilising a trained normalising flow: NeutraMALA and GflowMC. (a): Convergence of MCMC chains inferred from the observables of energy and magnetisation in the unimodal ($\theta = 1.0$) and bimodal phase ($\theta = 1.6$). Both samplers were roughly optimized: For NeutraMALA, the best performing target acceptance out of $\{0.2, 0.5, 0.75\}$ was chosen, for GflowMC, the optimal value of $d_{\mathrm{u}}$ for each observable was chosen, see fig. 3. (b): Wall time per MC step (500 chains evolved in parallel on identical GPU nodes).

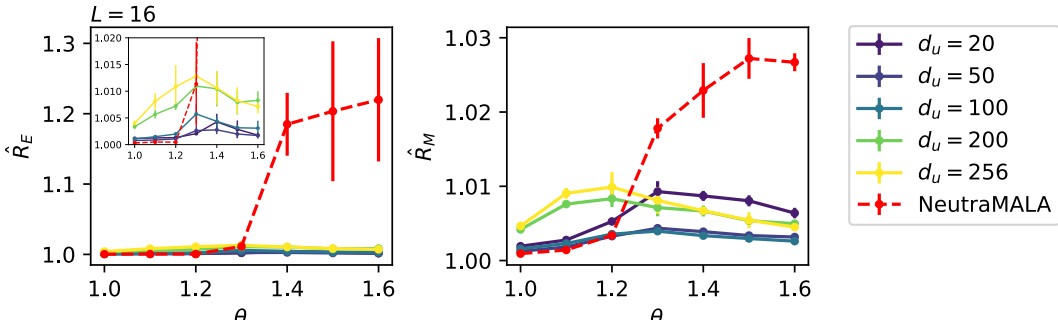

Figure 5: Statistics of GflowMC and NeutraMALA chains at different temperatures $\theta$. Different update subset sizes $d_u$ are shown for GflowMC. NeutraMALA was run with target acceptances 0.2, 0.5, 0.75, and in each case the best performing strategy is shown. The inset of the left panel shows the same data zoomed in. An average was taken over four indepdently trained flows, with the standard deviation indicated by error bars.

**Benchmark with competing NF-assisted sampler**  NeutraMALA is an appealing alternative to GflowMC: when full updates suffer from a large rejection rate, sampling the (approximately Gaussian) pull-back of the target distribution $\nu_\theta^*(z)$ with a MALA leaves a tunable step size to adjust the acceptance rate of the algorithm. Previous work finds NeutraMALA to be robust to flow quality, but not so well suited to multi-modal target distributions (33). We also see evidence of this in our experiments.

As shown on fig. 4(a), in the unimodal case of $\theta = 1.0$, NeutraMALA reaches slightly faster convergence than GflowMC as a function of the number of MC steps (similary for $\theta = 1.2$, not shown). In the bimodal case however, $\theta = 1.6$, convergence is much faster for GflowMC. For a fair comparison of the two schemes, computation time also needs to be taken into account. As a Langevin sampling scheme, NeutraMALA requires the computation of the gradient of $\nabla_z \log \nu_\theta^*(z)$, which can be obtained through automatic differentiation but is numerically costly. Comparing the average wall time to obtain the 20,000 steps MC chains, NeutraMALA is found to be by a factor of 3-4 slower than GflowMC (fig. 4(b)). Taking this into account, the latter algorithm is slightly superior even in the unimodal case. We conclude the comparison by reporting the GR statistics of full-chains of the two samplers across the phase transition, for $E$ and $M$ (see fig. 5). Here also, NeutraMALA gets worse as the distribution becomes bimodal. In contrast, the performance of GflowMC decreases slightly around the transition point, a behavior already reported by (11).

**Biasing schemes**  The natural extension of the partial updating scheme is to optimize the selection of latent coordinates for which updates are proposed. This is particularly relevant for the bimodal phase, where switching mode is the main challenge of the sampling procedure. To identify promising

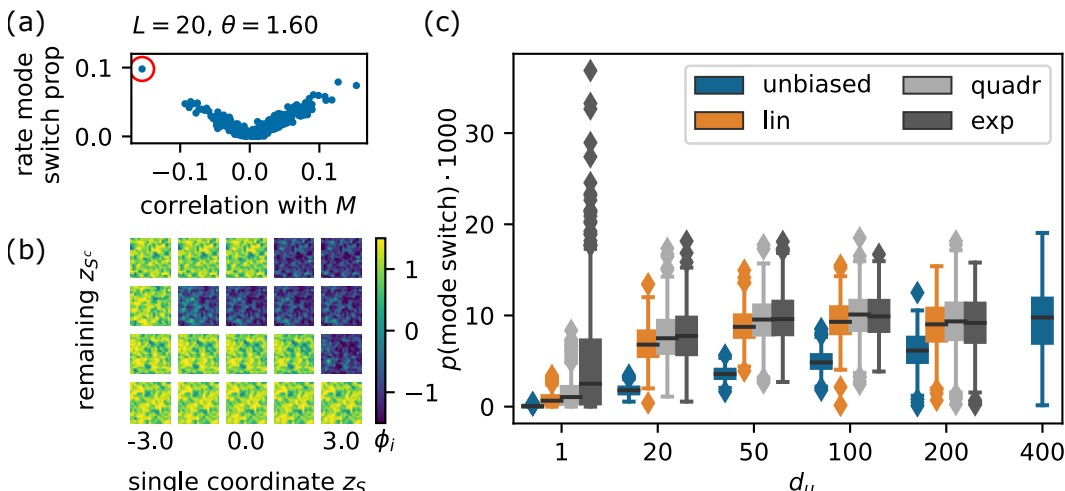

Figure 6: Mode switching abilities of an NF in the $\phi^4$ bimodal phase, $\theta = 1.6$, $L = 20$. (a) Mode switching rate for each latent coordinate $z_i$, determined by how often a mode change is proposed when resampling this coordinate keeping the remaining configuration fixed. (b) Configurations within a row differ by a single latent space coordinate $z_S \in \mathbb{R}$, chosen by having the largest $|c_i|$ (marked there in red in panel (a)) and varied in $[-3, 3]$. A single coordinate can induce a mode change in physical space. (c) Mode switching statistics of GflowMC with different update sizes and biasing schemes, chosen to promote the update proposal of coordinates correlated with the mode label.

latent coordinates $z_i$, we use the correlation coefficient $c_i$ of each $z_i$ with the magnetisation $M$. This choice can be justified when considering fig. 6(a), where the average rate of mode change when resampling this single coordinate is shown to be approximately proportional to $|c_i|$ for $L = 20$. We further illustrate this point in fig. 6(b), where the change of real space configurations is shown while just a single latent coordinate varied.

The coordinate selection is done choosing a non-flat $w \in \Delta^{d-1}$. For the assignment of the $w_i$, the coordinates $z_i$ are distributed into bins according to their value of $|c_i|$ in the interval $[0, 1]$. Then, each non-empty bin is assigned a weight proportionally to $f(|c|)$ distributed equally between all coordinates $z_i$ in the bin, given a biasing function $f(x)$. We ran experiments on three different schemes, with $f_{\text{lin}}(x) = x$, $f_{\text{quadr}}(x) = x^2$ and $f_{\text{exp}}(x) = \exp(50x)$ (an ad-hoc choice to achieve a strong biasing towards few coordinates).

As can be seen in fig. 6(c), the biasing schemes leads to more frequent mode switching at fixed $d_u$ while the most frequent mode switching on average is for the full update $d = d_u$. However, full updates come with a significant number of chains not switching mode at all. In contrast, the three biased strategies achieve comparable mode switching rate for partial update proposals, except for the $f_{\text{exp}}$ strategy which is more efficient than the others at small $d_u$. Nevertheless, we find that the overall convergence of $M$ is still best for the unbiased strategy at partial updates, presumably due to better within-mode exploration (fig. 7).

This motivates us to adapt our scheme towards a hybrid sampler and combine the flow update proposals with cheap MALA steps in real space. We run 10 MALA steps (with a target acceptance of 0.75) per GflowMC step. The results of this are shown in fig. 7. Comparing chains of equal length, the biasing schemes win in terms of the convergence of $E$, but for $M$, the partial update sampler without the MALA steps still performs best. However, it should also be taken into account that the GflowMC steps are more expensive than simple MALA steps (for 10 MALA steps per flow step, the overall run time differs by a factor of 3-4). We therefore also show results of the simple GflowMC sampler for chains which have a roughly comparable computation time. In this case, the hybrid samplers with biased strategy and partial updates actually outperform the pure GflowMC sampler. The most efficient strategy in this case therefore appears to be the combination of local MALA steps with partial updates biased towards mode-switching coordinates.

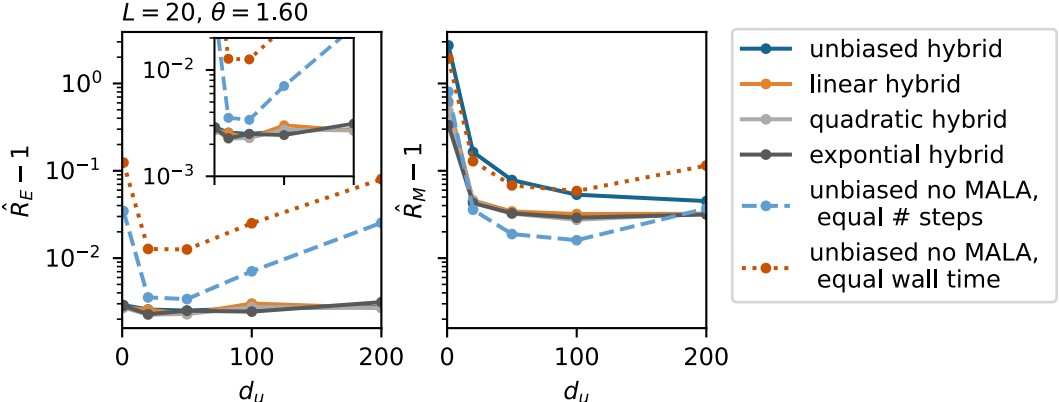

Figure 7: Comparison of the convergence for energy $E$ and magnetisation $M$ for the hybrid (global-local) sampler with biased partial update proposals for different schemes described in the main text. For comparison, we also show results for the simple GlowMC with equal chain length (unbiased no MALA) and the same for such chains shortened to have comparable wall time. The inset of the left panel shows the same data with zoomed-in y-axis.

## 4   Conclusion

In this study, we introduce GflowMC, a novel NF-enhanced sampler that leverages a Metropolis-within-Gibbs scheme implemented in the latent space. Our findings demonstrate that GflowMC effectively optimizes the trade-off between update size and acceptance rate, facilitating rapid decorrelation. Notably, we establish that the use of partial updates in the latent space enables mode-switches that are unattainable with either partial updates in coordinate space or local updates in the latent space. However, the number of accepted mode switches is not drastically improved by the biasing schemes overall. Biased schemes actually require to be combined with cost-efficient local space updates in coordinate space to show clear superiority. The exploration of such promising hybrid schemes—inspired by approaches like those in (19; 41)—is left for future work. Additionally, potential areas for further investigation include adaptive training of normalizing flows in tandem with GflowMC and the exploration of techniques for learning the weight vector $w_i$ (24).

## Acknowledgments and Disclosure of Funding

The authors thank Giuseppe Carlo for discussions that led to the idea of this work and Tony Lelièvre and Gabriel Stoltz for helpful discussions during the execution. The authors acknowledge funding from the Hi! Paris Center.

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

# A  Experimental settings

**Architectures**  We trained NFs for lattice sizes between $L = 8$ and $L = 24$ and for parameter values $\theta$ across the transition point (roughly at $\theta = 1.2$ in the thermodynamic limit $L \to \infty$ (35)). The flows were comprised of coupling layers (RealNVP and RQSpline), where we kept the number of layers constant and increased the width of the built-in neural networks as $L$ increases. The implementation was based on the Normflows package (42).

The flows were constructed of 12 RealNVP blocks and 2 RQSpline blocks with with alternating checkerboard masks. The neural networks inside the RealNVP layers had one hidden layer with $L \cdot L$ neurons. The RQSplines used 8 bins and neural networks with three hidden layers of size $L \cdot L$.

**Learning**  Training was done in the adaptive fashion introduced by (19), with 9 Langevin steps (target acceptance 0.75) per flowMC step for a batch size of 1,000. We used the Adam (43) optimizer (19) with a weight decay of $10^{-6}$ and a learning rate of $10^{-3}$ for the first 3,000 steps and $10^{-4}$ for the following 10,000.

