# OpenReview forum: "Optimizing Markov Chain Monte Carlo Convergence with Normalizing Flows and Gibbs Sampling"
_NeurIPS.cc/2023/Workshop/AI4Science — NeurIPS2023-AI4Science Poster_

### Official Review · Reviewer_2dnZ · 2023-10-22
**No novelty but still interesting for some workshop attendees**

**Rating:** 6
**Confidence:** 4

**Review:**

### Summary

The authors propose sampling unnormalized densities using Gibbs MCMC in the latent space of a pretrained normalizing flow. They provide experiments for a single 2D toy Boltzmann distribution. This system is extensively investigated

### Strengths

1. The paper tackles an important problem in sampling unnormalized densities which is a task that lies at the core of many scientific applications.

### Content concerns

1. The novelty is limited with MCMC in latent the latent space of a normalizing flow having been explored extensively before. It seems that the only novelty is performing Gibbs updates in the latent space which is well established. It seems to me that this (which MCMC scheme to use in latent space) is more of a hyperparameter choice than a technical contribution or insight.
2. Only a single toy system is used for experiments. This calls the generalizability of the obtained results into question.

### Presentation concerns

1.

### Recommendation

There is no novelty in the paper and the experimental evaluations are on a single toy system from which little meaningful conclusions can be drawn. I still think that highlighting the concept of latent space MCMC at this workshop again could be valuable to some attendees.

---

### Meta-Review · Area_Chair_DQ4a · 2023-10-27

**Recommendation:** Accept (Poster)
**Confidence:** 5

**Metareview:**

This work proposes a way to perform Gibbs sampling in the latent space of a previously trained Normalizing Flow model.
Despite the novelty of this work being limited, the paper is well structured and well written, thus constituting a pleasant read.

The authors perform numerical experiments on a common benchmark in the literature of Flow-based samplers for lattice field theory systems (despite using a rather unconventional parametrization for the action).

Upon carefully reading the paper myself, I found the authors' claim:
> "Using partial updates in the latent space, GflowMC improves mixing speed by finding the sweet spot between the amount of state refresh in the proposals and the rate of the proposals’ acceptance while keeping the ability to **mix between modes in multi-modal targets**".

 In light of this, I'd be very curious to see some discussions, in a future camera-ready version of the manuscript, on how GfloMC compares against standard flow-based sampling approaches using Reverse KL, in approximating multimodal distributions. Specifically, Ref [1], which may have escaped the authors' attention, deals explicitly with this problem for the same benchmark of the $\phi^4$ theory analysed in this paper. I'd be happy to see the authors comparing or commenting on this.

Furthermore, as a minor recommendation, I suggest the authors reorganize the references by order of appearance, as I find it pretty unusual to see the reference list ordered alphabetically.

In conclusion, I agree with the referee that, despite the limited novelty, it would still be interesting to have this paper accepted as it deals with a very hot topic that is worth having presented as a poster. Moreover, adding some discussions, as suggested above, would also give additional insights about the advantages, and potential novelty, of this work.

**References:**

- [1] [Nicoli, Kim A., et al. "Detecting and Mitigating Mode-Collapse for Flow-based Sampling of Lattice Field Theories." arXiv preprint arXiv:2302.14082 (2023).](https://arxiv.org/pdf/2302.14082)